# Genetic Population Flows of Southeast Spain Revealed by STR Analysis

María Saiz [1] , Christian Haarkötter [1] , Luis Javier Martinez-Gonzalez [2,*] , Juan Carlos Alvarez [1] and Jose Antonio Lorente [1,2]

1   Laboratory of Genetic Identification, Department of Legal Medicine, Toxicology and Physical, Faculty of Medicine, University of Granada, 18016 Granada, Spain
2   Centre for Genomics and Oncological Research, Pfizer-University of Granada-Andalusian Regional Government, 18016 Granada, Spain
*   Correspondence: luisjavier.martinez@genyo.es

**Abstract:** The former Kingdom of Granada, comprising the provinces of Granada, Málaga, and Almería (GMA), was once inhabited for over 700 years (711–1492 AD) by a North African population, which influenced its creation and establishment. The genetic data on 15 autosomal short tandem repeats (STRs) in 245 unrelated donor residents were examined in order to assess any possible admixture. As the two surnames in Spain follow an inheritance similar to the Y chromosome, both surnames of all 245 unrelated individuals were queried and annotated. The Spanish Statistics Office website was consulted to determine the regions with the highest frequency of individuals born bearing each surname. Further, several heraldry and lineage pages were examined to determine the historical origin of the surnames. By AMOVA and STRUCTURE analysis, the populations of the three provinces can be treated genetically as a single population. The analysis of allele frequencies and genetic distance demonstrated that the GMA population lay in the Spanish population group but was slightly more similar to the North African populations than the remainder of the Spanish populations. In addition, the surnames of most individuals originated in Northern and Central Spain, whereas most surnames had higher frequencies in Southern Spain. These results confirm that the GMA population shows no characteristics that reflect a greater genetic influence of North African people than the rest of the populations of the Iberian Peninsula. This feature is consistent with the historical data that African inhabitants were expelled or isolated during the repopulation of the region with Spaniards from Northern Spain. The knowledge of present populations and their genetic history is essential for better statistical results in kinship analyses.

**Keywords:** genetic variation; Kingdom of Granada; population genetics; Southern Spain; autosomal STRs; genetic legacy; distance analysis; structure

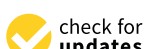



## 1. Introduction

### 1.1. Historical Aspects

The genetic legacy of the current population of the Iberian Peninsula was influenced by many invaders. For example, the Basque Country presents a genetic structure probably rooted in the Neolithic/Chalcolithic period (Günther et al. 2015) The invasion of the area that lies south of the Peninsula by North African populations for almost 800 years left a significant genetic footprint on this territory (Brion et al. 2003).

The present-day provinces of Granada, Málaga, and Almería, as well as portions of Cádiz, Jaén, Córdoba, and Sevilla, were all a part of the former Kingdom of Granada. Granada was the capital; it was one of the most flourishing cities in Europe during the 14th and 15th centuries (Chandler 1987).

Granada was established by the primitive Iberian tribes who founded *Iliberir*, which later became *Illiberis* under the Ancient Roman rule of Hispania. With the decline of the

Western Roman Empire in the 5th century, the Visigoths preserved the importance of the city and established it as a military stronghold for 300 years (415–711) invasions (Garzón 1980; Bueno 2004).

The first signal of African invaders was in 711 when Berbers arrived at the Iberian Peninsula and occupied the region of Granada, known as *Iliberir*, which concluded in the Caliphate of Córdoba. In 1013, the Zirid dynasty expanded their dominance over the region, expelling the Berbers, and founded *Ilbira* in 1025. The Zirid kingdom spread out to the entire territory of the Kingdom of Granada in order to avoid future invasions invasions (Garzón 1980; Bueno 2004). With the growth of the kingdom in 1090, a huge part of the Iberian Peninsula was reigned by the Zirid dynasty, known as *Al-Ándalus*. After the loss of the Battle of Navas de Tolosa in 1212, *Al-Ándalus* was reduced to the Nasrid Kingdom of Granada, to what today corresponds to Granada, Málaga, Almería, and some parts of Córdoba, Sevilla Jaén, and Cádiz. The Nasrid dynasty was the longest-lasting Muslim dynasty in the Iberian Peninsula. Finally, the Kingdom of Granada came to an end with the conquest of the city of Granada by the Catholic Monarchs Ferdinand II and Isabel I in 1492 invasions (Garzón 1980; Bueno 2004).

Although Muslims signed capitulations to adhere to the religion of the kingdom, they were forced to convert to Christianity or emigrate. Once the Moriscos' belongings were expropriated, it became imperative to repopulate the region with new inhabitants from several regions of the Peninsula. The repopulation began in 1571 and persisted until 1595, a total of 12,546 families repopulated 270 areas (Bueno 2004). On 9 December 1609, Philip III signed the expulsion order of all Moriscos from Spain (Garzón 1980; Bueno 2004). In 1833, after 314 years of existence and with the separation of the provinces of Almería and Málaga, the former Kingdom of Granada finished invasions (Garzón 1980; Bueno 2004).

Thus, during the creation of the Kingdom of Granada and during its existence, people of various religions and regions cohabited. The coexistence of Muslims, Jews, and Christians resulted in a pluralism that is evident in the architecture, culture, and folklore of the present-day cities of Granada, Málaga, and Almería.

### 1.2. Genetic Aspects

Studies based on ALU sequences discovered sub-Saharan gene traces in north Mediterranean populations, suggesting continuous interactions between both coasts coasts (González-Pérez et al. 2010). The fact that genetic traits and certain specific haplotypes have been found along the north coast of the Mediterranean supports the idea that gene flow in this region is related to the first trans-Mediterranean sailings and stayed homogeneous while trade slave lasted into the late 17th century, rather than being a result of Islamic expansion (S. VII to S. XV) coasts (González-Pérez et al. 2010). The analysis of genome-wide SNP data from over 2000 individuals has allowed the characterization of broad clinal patterns of recent gene flow between Europe and Africa that have a considerable effect on the genetic diversity of European populations, especially in the southwest European populations (Botigué et al. 2013). Though, contrary to what might be expected based on historical data, a gradient exists from south to north of North African genetic influence; most genetic influence is found in Galicia and northern Castilla (>20%). The main North African gene frequencies gradient is observed between the west and east, where smaller proportions are detected. In addition, recent studies based on autosomal SNPs (Bycroft et al. 2019) and Y chromosome lineages (Rey-González et al. 2017) show that the Andalusian population does not particularly group with North African populations more than other Iberian populations (Larmuseau and Ottoni 2018). After the Reconquest, the Moors were disseminated homogeneously in the Peninsula, but their final expulsion in 1609 was much more effective in some provinces of Spain; Valencia, and Western Andalusia, while in Galicia and Extremadura, the population dispersed and integrated into the society (Adams et al. 2008).

Kinship analysis in forensics is based on the calculation of several kinship indices and likelihood ratios. These statistics are calculated based on allele frequency data for the

studied set of STR markers from the population to which the individuals belong. The use of the appropriate allele frequency population data is fundamental to assure the inference of relationships between two individuals.

The main objective of this study was to detail the genetic variations in the populations of Granada, Málaga, and Almería by examining 15 short tandem repeats and, thus, establish their phylogenetic positions with respect to those of other European and North African populations in the literature and to determine whether the possible North African genetic influence is higher than in the rest of Iberian Peninsula populations and to create specific allele frequency population data for the Southeast Spanish population.

## 2. Results

### *2.1. Autosomal STRs Allele Frequencies*

The distribution of the observed allele frequencies of the 15 STR loci is shown in Table 1, which also lists the power of discrimination (PD), power of exclusion (PE), and observed (Ho) and expected (Ht) heterozygosis. The most informative markers were D18S51 and FGA; the least descriptive marker was TPOX. The combined discriminatory power and the combined exclusion power for the entire population were 1—$5.55362\cdot10^{-18}$ and 99.9997%, respectively.

**Table 1.** Allele frequencies of the Identifiler STR loci in the GMA population sample. Forensic summary statistics, observed and expected heterozygosity, and deviation from Hardy–Weinberg equilibrium (HWE).

| Allele | D8S1179 | D21S11 | D7S820 | CSF1PO | D3S1358 | THO1 | D13S317 | D16S739 | D2S1338 | D19S433 | VWA | TPOX | D18S51 | D5S818 | FGA |
|---|---|---|---|---|---|---|---|---|---|---|---|---|---|---|---|
| 6 | | | 0.004 | | | 0.218 | | | | | | | | | |
| 6.3 | | | | | | 0.002 | | | | | | | | | |
| 7 | | | 0.033 | 0.002 | | 0.159 | | | | | | | | | |
| 7.3 | | | | | | 0.002 | | | | | | | | | |
| 8 | 0.006 | | 0.131 | 0.008 | | 0.147 | 0.161 | 0.022 | | | | 0.484 | | 0.004 | |
| 9 | 0.018 | | 0.131 | 0.014 | | 0.188 | 0.061 | 0.118 | | | | 0.135 | | 0.022 | |
| 9.3 | | | | | | 0.271 | | | | | | | | | |
| 10 | 0.086 | | 0.280 | 0.291 | | 0.010 | 0.041 | 0.047 | | | | 0.080 | 0.006 | 0.080 | |
| 11 | 0.096 | | 0.202 | 0.319 | | 0.002 | 0.302 | 0.251 | | 0.006 | | 0.278 | 0.012 | 0.354 | |
| 12 | 0.118 | | 0.171 | 0.297 | | | 0.290 | 0.339 | 0.109 | | 0.002 | 0.024 | 0.182 | 0.327 | |
| 12.2 | | | | | | | | | | 0.002 | | | | | |
| 13 | 0.294 | | 0.047 | 0.056 | 0.004 | | 0.090 | 0.190 | | 0.246 | 0.002 | | 0.154 | 0.197 | |
| 13.2 | | | | | | | | | | 0.008 | | | | | |
| 14 | 0.233 | | 0.002 | 0.012 | 0.076 | | 0.051 | 0.031 | | 0.371 | 0.116 | | 0.159 | 0.014 | |
| 14.2 | | | | | | | | | | 0.016 | | | | | |
| 15 | 0.127 | | | | 0.300 | | 0.004 | | | 0.145 | 0.129 | | 0.129 | 0.002 | |
| 15.2 | | | | | | | | | | 0.035 | | | | | |
| 16 | 0.020 | | | | 0.235 | | | 0.002 | 0.045 | 0.041 | | 0.282 | 0.117 | | |
| 16.2 | | | | | | | | | | 0.012 | | | | | |
| 17 | 0.002 | | | | 0.171 | | | | 0.247 | 0.006 | | 0.229 | 0.104 | | 0.002 |
| 17.2 | | | | | | | | | | 0.002 | | | | | |
| 18 | | | | | 0.200 | | | | 0.086 | | 0.169 | | 0.055 | | 0.016 |
| 19 | | | | | 0.014 | | | | 0.114 | | 0.061 | | 0.045 | | 0.063 |
| 20 | | | | | | | | | 0.147 | | 0.010 | | 0.016 | | 0.129 |
| 21 | | | | | | | | | 0.039 | | | | 0.016 | | 0.211 |
| 22 | | | | | | | | | 0.029 | | | | 0.004 | | 0.134 |
| 22.2 | | | | | | | | | | | | | | | 0.004 |
| **Allele** | **D8S1179** | **D21S11** | **D7S820** | **CSF1PO** | **D3S1358** | **THO1** | **D13S317** | **D16S739** | **D2S1338** | **D19S433** | **VWA** | **TPOX** | **D18S51** | **D5S818** | **FGA** |
| 23 | | | | | | | | | 0.098 | | | | | | 0.146 |
| 23.2 | | | | | | | | | | | | | | | 0.004 |
| 24 | | | | | | | | | 0.098 | | | | | | 0.153 |
| 24.2 | | | | | | | | | | | | | | | 0.002 |
| 25 | | | | | | | | | 0.086 | | | | | | 0.090 |
| 26 | | 0.004 | | | | | | | 0.008 | | | | | | 0.033 |
| 26.2 | | 0.002 | | | | | | | | | | | | | |
| 27 | | 0.018 | | | | | | | 0.004 | | | | | | 0.014 |
| 28 | | 0.113 | | | | | | | | | | | | | 0.002 |
| 28.3 | | 0.002 | | | | | | | | | | | | | |
| 29 | | 0.192 | | | | | | | | | | | | | |
| 30 | | 0.307 | | | | | | | | | | | | | |
| 30.2 | | 0.027 | | | | | | | | | | | | | |
| 31 | | 0.061 | | | | | | | | | | | | | |
| 31.2 | | 0.102 | | | | | | | | | | | | | |
| 32 | | 0.010 | | | | | | | | | | | | | |
| 32.2 | | 0.122 | | | | | | | | | | | | | |
| 33 | | 0.002 | | | | | | | | | | | | | |

**Table 1.** *Cont.*

| Locus | D8S1179 | D21S11 | D7S820 | CSF1PO | D3S1358 | THO1 | D13S317 | D16S739 | D2S1338 | D19S433 | VWA | TPOX | D18S51 | D5S818 | FGA |
|---|---|---|---|---|---|---|---|---|---|---|---|---|---|---|---|
| *33.2* | | 0.029 | | | | | | | | | | | | | |
| *34.2* | | 0.004 | | | | | | | | | | | | | |
| *35* | | 0.004 | | | | | | | | | | | | | |
| PD | 0.940 | 0.947 | 0.939 | 0.866 | 0.916 | 0.926 | 0.924 | 0.915 | 0.965 | 0.912 | 0.934 | 0.832 | 0.966 | 0.875 | 0.964 |
| PE | 0.577 | 0.677 | 0.526 | 0.449 | 0.562 | 0.562 | 0.540 | 0.484 | 0.725 | 0.474 | 0.491 | 0.332 | 0.766 | 0.451 | 0.708 |
| PIC | 0.790 | 0.670 | 0.790 | 0.670 | 0.740 | 0.720 | 0.750 | 0.730 | 0.850 | 0.680 | 0.780 | 0.610 | 0.860 | 0.670 | 0.804 |
| $H_{obs}$ | 0.788 | 0.841 | 0.759 | 0.714 | 0.780 | 0.780 | 0.767 | 0.735 | 0.865 | 0.730 | 0.739 | 0.633 | 0.886 | 0.714 | 0.857 |
| $H_{exp}$ | 0.814 | 0.827 | 0.816 | 0.725 | 0.781 | 0.798 | 0.784 | 0.770 | 0.868 | 0.767 | 0.807 | 0.665 | 0.873 | 0.725 | 0.865 |
| p | 0.633 | 0.193 | 0.134 | 0.079 | 0.904 | 0.585 | 0.634 | 0.613 | 0.594 | 0.223 | 0.093 | 0.124 | 0.632 | 0.425 | 0.502 |

$H_{obs}$, observed heterozygosity; $H_{exp}$, expected heterozygosity; PD, power of discrimination; PE, power of exclusion; PIC, polymorphism information content; P, HWE, Fisher's exact test *p*-value executed with 100,000 steps in the Markov chain and 10,000 dememorization steps. None of the markers deviated from the Hardy–Weinberg equilibrium, and all had normal values of heterozygosis but no signs of linkage between loci.

### 2.2. Population Substructure

The hypothesis of a disparate genetic structure between the three provinces was tested by AMOVA (Supplementary Table S1). No significant genetic substructure was detected between subpopulations (Supplementary Table S2); only 2.32% of the variation was observed among the populations (*p*-value 0.00238).

These results were confirmed by STRUCTURE analysis. No evidence of any significant genetic substructure was observed between the clusters of the GMA population. The model with the highest posterior probability value was K = 1 (ln P[D] = −12,981.88), compared with K = 3 (ln P[D] = −13,300.52), implying that the genetic data favor a single cluster for the three subpopulations.

### 2.3. Population Cross-Comparisons

Correspondence analysis was performed in Statistica v9.1 for South Europe and North African populations (Figure 1). To simplify the interpretation of the data, the figure omits markers' data and shows only population results. Two central groups could be detected in the figure: the Spanish populations in red and the North African populations in blue. The GMA population lay in the Spanish group, next to the Catalan and Andalusian populations.

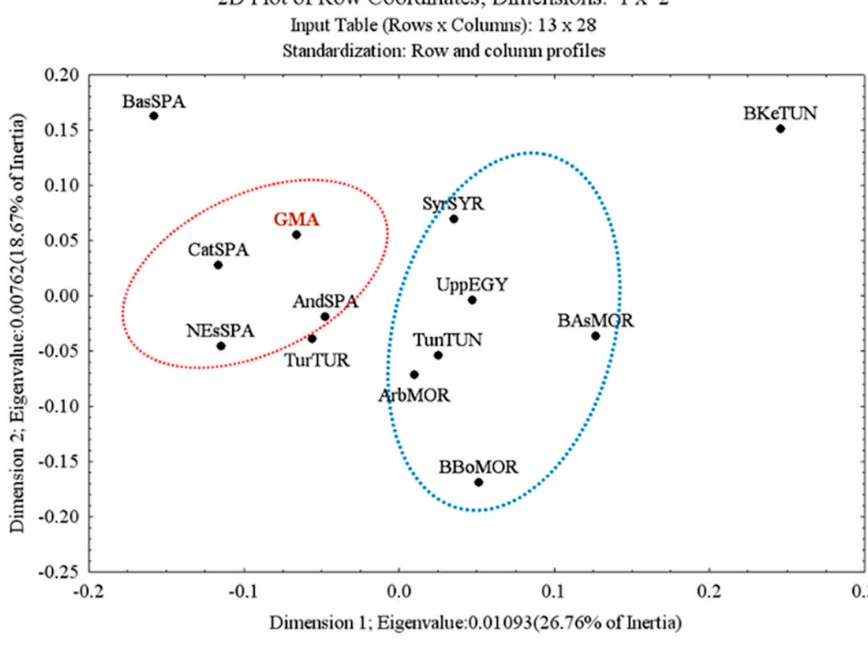

**Figure 1.** Two-dimensional plot of correlation analysis to assess the association between allele frequencies and populations.

The Nei, Reynold, and Cavalli-Sforza genetic distances were calculated with the *Gendist* application to decipher the genetic relationships that appeared, based on the genome-wide autosomal markers, between the GMA populations and 12 other populations from the literature (Supplementary Table S3) by considering the allele frequencies of the 13 STR CODIS loci. Nonmetric multidimensional scaling (MDS) was performed using IBM SPSS Statistics 20 to graphically plot the genetic distance matrix (Figure 2). Dimension 1 clearly separated North African populations (Arabs from Morocco, Berbers from Bouhria, Berbers from Asni, and Berbers from Kesra, Tunisia, Upper Egypt, Turkey, and Syria), located in the negative area, from South European populations (Basque Country, Catalonia, northeast Spain, and general population from all Andalusia and GMA from Spain), located in the positive area.

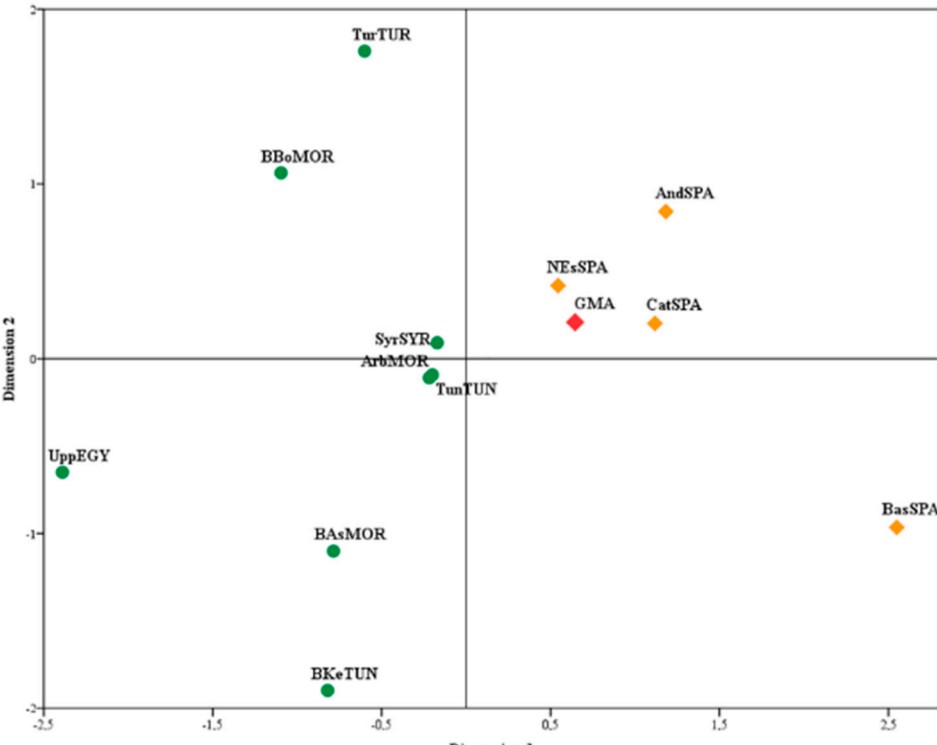

**Figure 2.** Multidimensional scaling plot applied to the Nei (Fst) genetic distance; stress value 0.19979, RSQ = 0.83546.

To complement these analyses, STRUCTURE was used to determine whether any broad genetic structure existed between the GMA population and the worldwide population dataset. The model with the highest posterior probability value was at K = 5 ln P[D]) = −94172.89 and Delta K = 20.4368 (Supplementary Table S4). In this analysis, whereas nearly all individuals showed membership in only predominant clusters that corresponded to geographical affiliation, the Moroccan population showed contributions by several clusters. For example, all European samples exhibited a distinct "European" cluster (Figure 3, in red), Somalis formed an "East African" cluster (green), and South Africans were represented by an East African component and a South African component (purple). Libya represented its own cluster (light blue) due to the high consanguinity rates for this population (Elmrghni et al. 2012). The correlation between clusters and populations was not distinct for the Moroccan sample; however, individuals showed partial membership in several clusters (European and sub-Saharan) due to the ethnic admixture that built this population (Arabs, Berbers, and Sahrawi) (Bouabdellah et al. 2008). These results implicate the existence of identities that do not have any geographic, linguistic, or ethnic affiliation. The GMA population distinctly belongs to the European cluster without an African component.

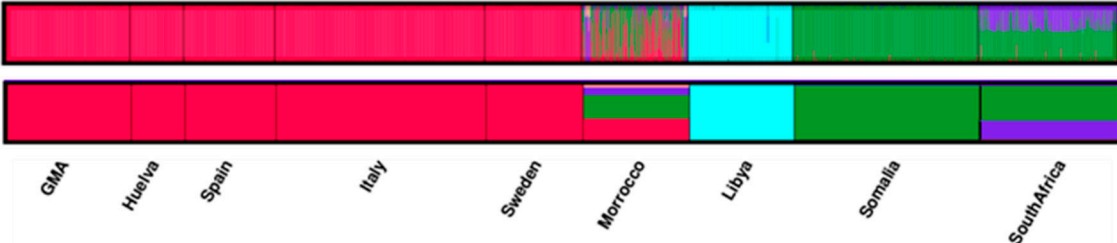

**Figure 3.** Structure analysis at K = 5 for nine populations. Average individual assignments to clusters for structure analyses. Each individual is represented by a thin vertical line, which is partitioned into K colored segments that represent the estimated membership fractions in each K cluster (upper figure). Average population assignment to clusters for structure analysis (bottom figure).

*2.4. Surnames*

In the sample of 245 individuals, 266 different surnames were recorded, 197 of which were singletons; the remaining 69 surnames ranged in absolute frequency from 2 to 23 (Supplementary Table S5).

In Spain, there are 26,223 surnames with a frequency of over 20, according to the 2021 census (Instituto Nacional de Estadistica, www.ine.es, accessed on 11 April 2023). The most frequent surname in the population is García (3.71%). Figure 4a shows the distribution of the nonsingleton surnames in the sample and their respective values in the entirety of Spain and their weighted averages in Granada, Málaga, and Almería—the three frequency distributions were similar. On a national scale, 29 of those repeated surnames occurred at frequencies of lower than 0.001, and those of 5 surnames were <0.0001. On a provincial scale, the corresponding values were 19 and 2, respectively.

The population was differentiated into eight subgroups based on their geographical location and historical origin (Figure 5). The surnames of most individuals originated in Northern and Central Spain. A total of 37% of the surnames originated from the center of Spain, where the crown of Castilla reigned; 35% came from the region that lies north of Spain, where an important Celtic cultural influence can be observed; and 10% was derived from Aragon versus 4% from Cataluña and 7% from Andalucía, the former Kingdom of Granada (Figure 5a). However, considering the regions with the highest frequency of individuals born bearing each surname, 37% of the surnames are most frequently seen in Andalucía, compared with 22% from Castilla, 20% from the north of Spain, and 10% from other regions, such as the Canary Islands (Figure 5b).

Finally, 10 of the 266 surnames in the sample had an Arabic etymological origin (8 first surnames and 4 s surnames, 2 of them in both the first and second names), 8 of which were unique to each group. The surnames in both the first and second surnames were as follows: Simon in four individuals (three first surnames and one second surname) and Medina in two individuals (one in each surname). Figure 3b shows the distribution of the 10 surnames with Arabic etymological origin in the sample and their respective values in the entirety of Spain and in Granada, Málaga, and Almería—the five frequency distributions were similar. In addition, 7 of the 10 individuals were born in the province with the highest frequencies for each surname.

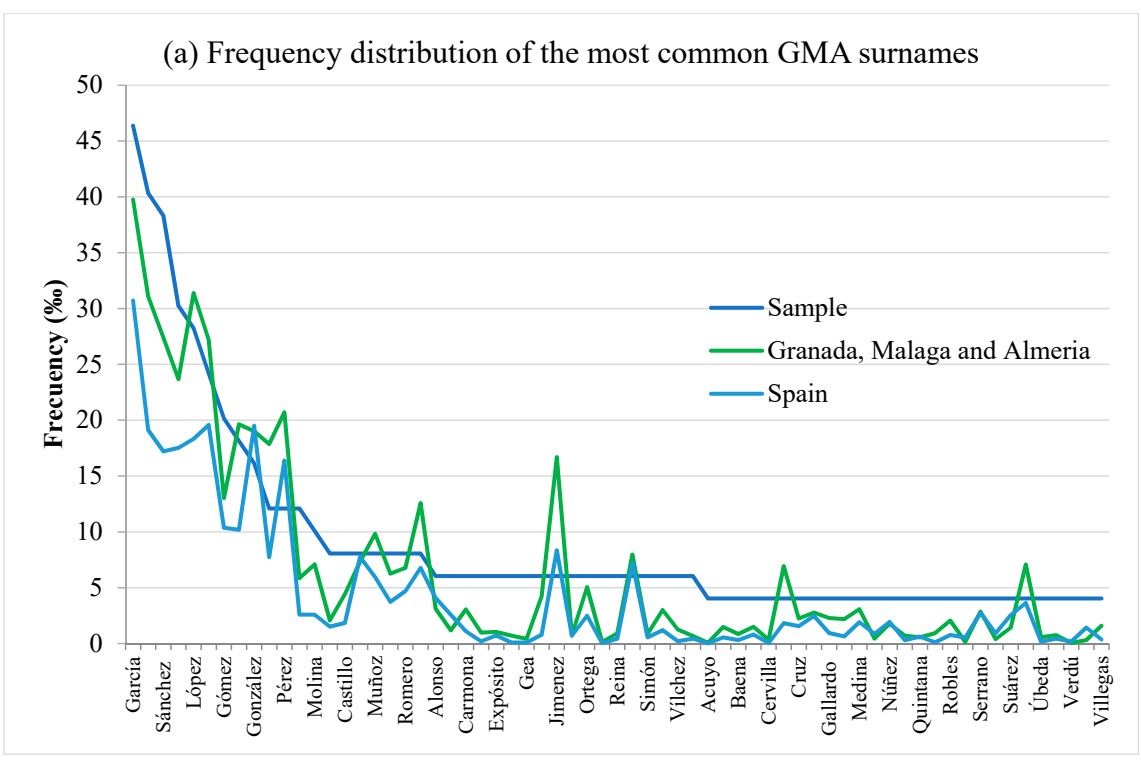

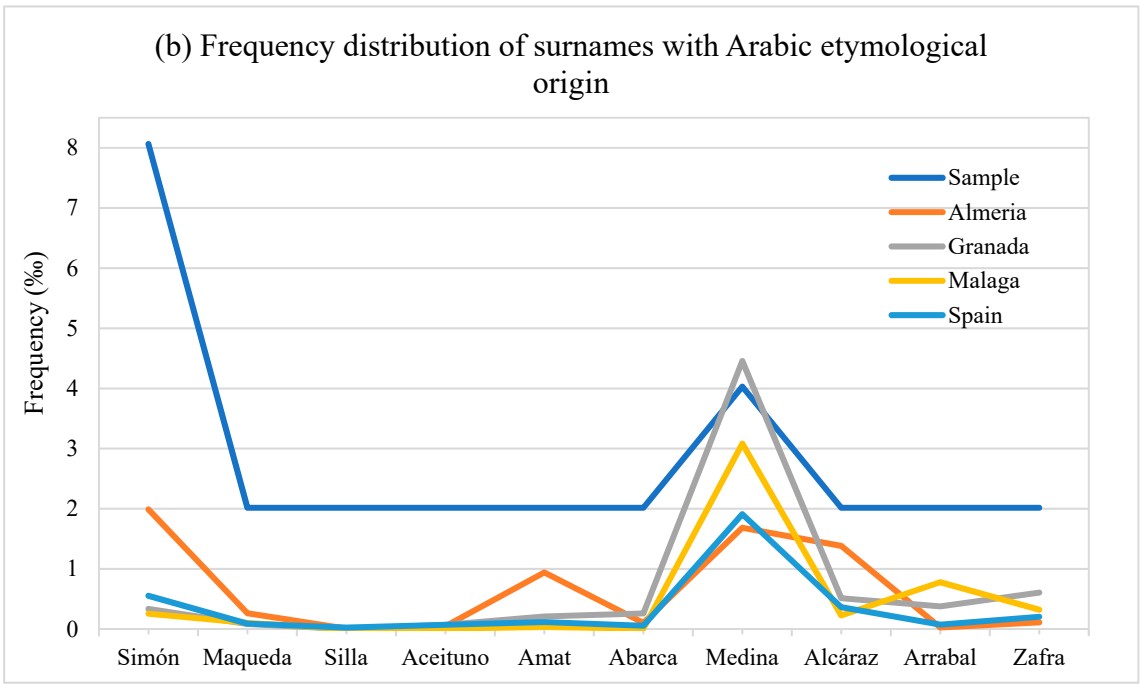

**Figure 4.** (**a**) Frequency distribution of the most common GMA surnames, (**b**) frequency distribution of surnames with Arabic etymological origin. Values in GMA sample were compared with those in the weighted average of Granada, Málaga, and Almería provinces and Spain (Statistics of the Continuous Census; 1 January 2021; INE).

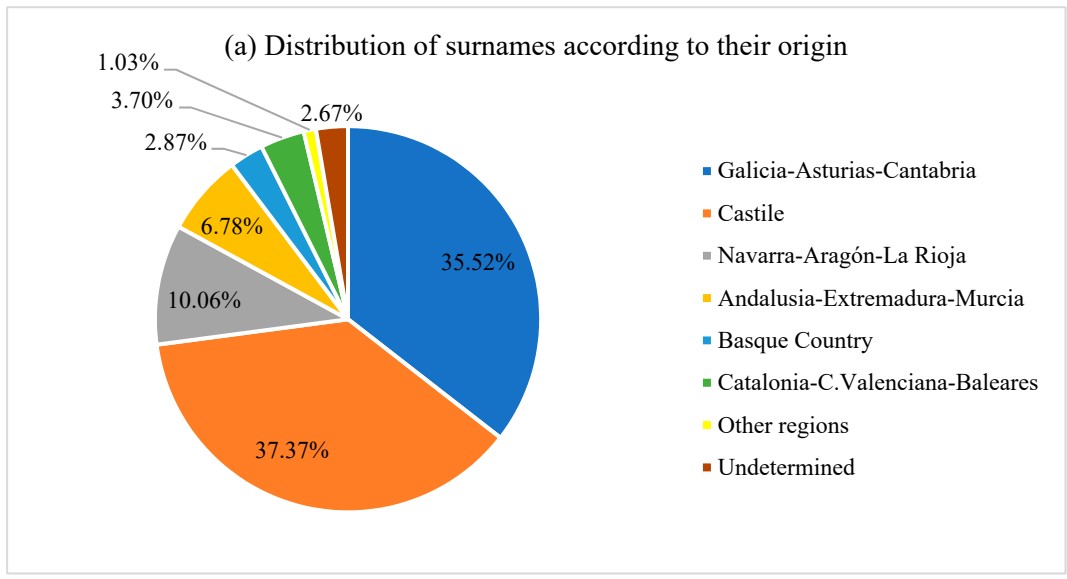

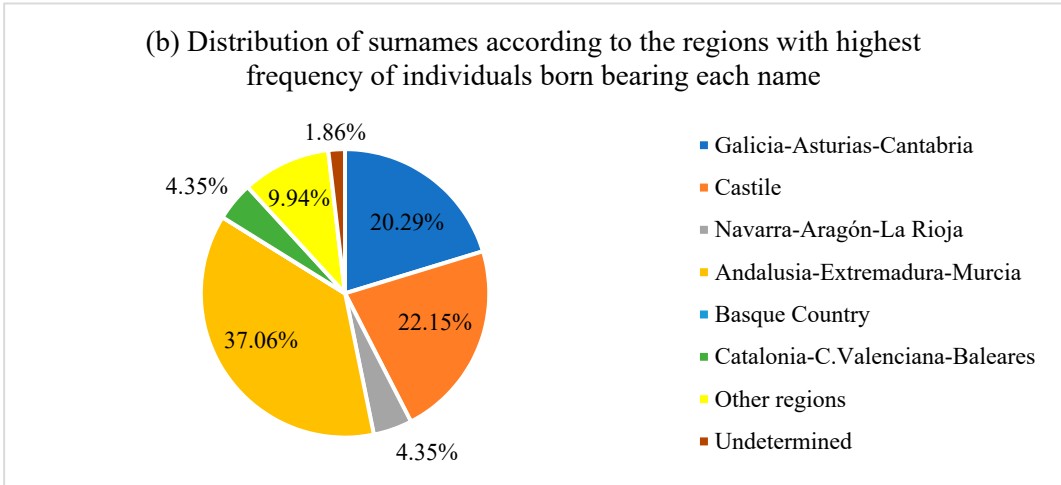

**Figure 5.** Distribution of the surnames from the population of the study according to (**a**) their origin and (**b**) the regions with the highest frequency of individuals born bearing each surname.

## 3. Discussion

Many groups have studied the genetic relationships between North African and South European people (Capelli et al. 2009; Plaza et al. 2003) and those in the Iberian Peninsula (Brion et al. 2003; Bycroft et al. 2019; Adams et al. 2008; Pérez-Lezaun et al. 2000; Bertranpetit and Cavalli-Sforza 1991; Bosch et al. 2001) to determine the genetic legacy that remains in present-day populations. To detail the proposed existence of genetic relationships between South Iberian populations and North African invaders, the population of the provinces that comprised the former Kingdom of Granada was analyzed and compared with other Spanish and North African populations.

Prior to this comparison, to treat the samples as a single population or three independent groups, AMOVA and STRUCTURE analysis were performed. A comprehensive geographical coverage of the three current provinces was performed to select the samples included in this study, comprising both coastal and inland towns as far as a proportional distribution of samples from the capital cities of the provinces. No subdivisions were seen in terms of the geographical origin of the samples, as evidenced by the STRUCTURE analysis. Samples could not be grouped into more than one cluster, and few variations between populations were observed by AMOVA. This finding is consistent with historical

and sociocultural expectations based on the shared origin of these populations with regard to their geographical proximity.

In the first level of the comparison, based on the allele frequencies, the GMA population fell within the Spanish populations (Figure 1). In addition, the population from Basque Country lay farther from the rest of the Spanish population due to the differences between the D13S317 (allele 8), TPOX (allele 12), and TH01 markers (allele 9.3); data in Supplementary Table S6. The use of a large number of SNPs confirms that Basques are differentiated from other European populations (Rodríguez-Ezpeleta et al. 2010), confirming its position in the correlation analysis.

Similar results were obtained in the study of genetic distances by MDS. Two clusters were observed, coinciding with the geographical distribution of the populations. Dimension 1 clearly separated the North African from Spanish populations. The GMA population clustered with other Spanish populations (Figure 2).

Based on the study of the 15 autosomal STRs, in the distance analysis, the North African populations had little influence on the GMA population (Figure 2), not higher than in other Iberian Peninsula populations. It is difficult to fathom that few African components survived despite 700 years of occupation. The similarity between the GMA and European populations—specifically, the Spanish populations—rendered the identification of the differences between them difficult. Further, the STRUCTURE analyses confirmed these results (Figure 3). The similarity might be attributed to the lack of genetic interaction between the Muslim population that inhabited the territory and the Spanish conquerors. Historical data indicate that the Muslim people who inhabited the Kingdom of Granada were expelled or isolated and that few were Christianized and remained in the region. In addition, after occupation of the city of Granada by the Spaniards in 1492, people were taken from the north to inhabit the region, thus isolating the Muslims further. These data are supported by the results on the origin of the surnames of the individuals, wherein the surnames that originated in the north or center of Spain are more common today in the south.

North African ancestry in Europe and, in particular, in the Iberian Peninsula has been broadly studied. Genome-wide SNP data from over 2000 North African and European individuals show that recent North African ancestry is highest in Southwestern Europe, with levels rising to 20% (Botigué et al. 2013). Studies based on autosomal single nucleotide polymorphisms in populations of the Iberian Peninsula show that North African ancestry does not reflect proximity to North Africa or even regions under more extended Muslim control; the highest amounts of North African ancestry found within Iberia are in the west (Bycroft et al. 2019), supporting previous studies based on Y chromosome binary markers. These studies determined that the Islamic rule of Spain left only a minor contribution to the current Iberian Y chromosome pool (Bosch et al. 2000), such that the highest proportions of North African ancestry are found in Galicia and Northwest Castile (Adams et al. 2008). Similar results have been observed in a detailed analysis of Y chromosome STR markers in the same population (Saiz et al. 2019). In addition, recent mitochondrial DNA analysis based on 7611 control region sequences revealed that typical sub-Saharan and North African lineages are slightly more prevalent in South Iberia, although at low frequencies (Barral-Arca et al. 2016).

Furthermore, genomic data from 45 individuals dated between the 3rd and the 16th centuries reveal that current populations from the south of the Iberian Peninsula hold less North African ancestry than the ancient Muslim burials, reflecting the expulsion of Moriscos and repopulation from the north (Olalde et al. 2019).

Even though the microsatellites used in this study were selected for forensic studies because of their high degree of variation within populations, their levels of interpopulation variation are relatively low but sufficient to assess recent genetic relationships between populations. This makes them a useful tool in population genetic studies based on migratory movements that occurred in the last centuries (Gaibar et al. 2012; Dahbi et al. 2023). However, studies with other lineage polymorphisms, such as mitochondrial DNA or Y

chromosome polymorphisms, are necessary to fully support the results obtained with autosomal markers.

In Spain, the use of surnames became widespread among the Christian population in the 10th century but did not expand throughout the population until the 12th century. However, until the Council of Trent (1545–1563), informal and lax rules on surnames were established. The introduction of surnames during the Middle Ages coincides with the reconquest of the territory that was under Muslim rule.

The wide range of surnames in the GMA samples is supported by the history of this region. Historical data indicate that the Moors who inhabited the Kingdom of Granada were expelled and isolated. Few of them converted to Christianity and remained in the region; those who did were known as new Christians. During this period, Muslims and Jews adopted Christian surnames, as well as the male inheritance system of these names. Later, after the occupation of the city of Granada by the Spaniards in 1492, the Moors were relegated to the zone of the Alpujarra until 1570, from where they were expelled. All of these regions were repopulated with people from the north and center of the Peninsula. There was no contact between new Christians and old Christians. In 1609, all Moors and new Christians were expelled from the Iberian Peninsula to North Africa.

The tremendous isolation of the Moriscos and the little contact between them and the new settlers are reflected in our results on surnames. These patterns explain how most surnames had a Castilian or Galician origin, whereas most surnames had higher frequencies in Southern Spain. A total of 12.57% of surnames with the highest number of individuals who were born in the south of the Peninsula were Galician, Asturian, or Cantabrian in origin (Figure 5b). Among them, there are such names as Carmona, Cubero, Ferrón, Montes, Padial, Rojas, and Santiago; i.e., nowadays surname Padial is mostly represented in the province of Granada, 11.04%, but it has no representation in Galicia. Conversely, 18.03% of surnames with the highest number of individuals who were born in the south of the Peninsula came from the center of the Peninsula, Castilla León, and Castilla la Mancha, such as Burgos, Castillo, Domínguez, Guerrero, León, and Romero (Figure 5b). Finally, 7.65% of surnames with the highest number of individuals who were born in the south of the Peninsula were Navarrese–Aragonese in origin—e.g., Aragon, Cortés, Navas, and Soto (Figure 5b). Although many languages have historically been spoken in the Iberian Peninsula—Castilian, Portuguese, Galician, Basque, Catalan, Arabic, and Hebrew, giving rise to certain characteristic surnames—most of the Spanish population has surnames of Castilian–Leones origin, which predominate the entire Spanish province (Calderón et al. 2015).

## 4. Materials and Methods

### 4.1. Population Sample

Buccal cell swabs were collected from 245 unrelated adult males and females in Granada (94), Málaga (72), and Almería (79), spanning at least three generations, and all four grandparents were born in the sampling area. The origins of the samples are shown in Supplementary Figure S1.

### 4.2. Autosomal STR Typing

Genomic DNA was isolated with phenol/chloroform/isoamyl alcohol extraction and proteinase K digestion and purified on Amicon 100 (Millipore). The extracted DNA was quantified on a 0.8% agarose gel. The samples were amplified using the AmpF*l*STR Identifiler and AmpF*l*STR Identifiler Plus kits (Applied Biosystems, Foster City, CA, USA) under manufacturer's recommendations (Applied Biosystems 2015). Alleles were separated and detected on an Applied Biosystems ABI 310 genetic analyzer. Fragment sizes were analyzed using GeneMapper ID-X v1.1 (Applied Biosystems, Foster City, CA, USA). The alleles were named according to the number of repeated units based on the sequenced allelic ladder (ISFG recommendations) (Bär et al. 1997).

### 4.3. Statistical Analysis

Allele frequencies, heterozygosity (H), polymorphism information content (PIC), power of discrimination (PD), power of exclusion (PE), matching probability (MP), and typical paternity index (TPI) were calculated for each locus using STRAF 1.0.5 (Gouy and Zieger 2017). Hardy–Weinberg proportion and linkage disequilibrium between pairs of loci were tested in Arlequin v3.5.1.2 (Excoffier and Lischer 2010) by exact test based on 10,000 shuffling experiments, and for detecting disequilibrium between STR loci, an inter-class correlation criterion for 2-locus associations was used. Analysis of molecular variance (AMOVA) was performed with Arlequin v3.5.1.3. AMOVA measures the proportion of variance within and between populations or groups of populations. Genetic differentiation and genetic distance (Fst) coefficients for the populations of Granada, Málaga, and Almería were calculated using Arlequin v3.5.1.3 (Excoffier and Lischer 2010). Published allelic frequency data and genetic profiles from several populations were compiled. Additional information on these populations is summarized in Supplementary Table S1.

Autosomal STR allele frequencies were used to calculate genetic distances with the *Gendist* application included in the *Phylip v3.69* informatics package (Felsenstein 2004). To generate a more appropriate representation of the distances, genetic distances (the Nei, Reynold, and Cavalli-Sforza genetic distance matrices) were summarized graphically by nonmetric multidimensional scaling (NM-MDS) (Kruskal 1964) using *IBM SPSS Statistics 20* (IBM Corp., Armonk, NY, USA). Correspondence analysis was performed with *Statistica v9.1* (Statsoft Inc., Tulsa, OH, USA) to understand the association between allele frequencies and populations. Two markers were eliminated due to a lack of data in certain populations (D2S1338 and D19S433).

*STRUCTURE v2.3.1* (Falush et al. 2007; Hubisz et al. 2009) was used to implement the estimation of the proportions of individual ancestries. Replicate runs of STRUCTURE using different burn-in periods and interactions were performed. For all simulations and calculations, no-admixture and admixture models were assumed, including prior population information, and the correlation between groups was determined with allele frequencies. The estimations were calculated with a burn-in period of 50,000 interactions, followed by an additional 100,000 interactions (K = 1 to 10), and a model of independent allele frequencies was specified. Structure analysis was replicated 10 times for each choice, and posterior probabilities for each *K* were computed for each set of runs. The 10 replicates for each choice of *K* were evaluated using CLUMPP (Jakobsson and Rosenberg 2007). The combined clustering result was visualized with DISTRUCT 1.1 (Rosenberg 2004).

### 4.4. Surname Study

As surnames in Spain follow an inheritance similar to the Y chromosome, both surnames of all 245 unrelated individuals were queried and annotated. The Spanish Statistics Office website (www.ine.es/en, accessed on 11 April 2023) was consulted to determine the regions with the highest frequency of individuals born bearing each surname. Further, several heraldry and lineage pages were examined to determine the historical origin of the surnames. The population was divided into eight subgroups and classified by surname origin and the birthplace of the bearer. Surnames were compared with an available list of Spanish surnames of Arab origin (Calvo Baeza 1990).

## 5. Conclusions

The former Kingdom of Granada comprised the current territories of Granada, Málaga, and Almería, behaving as a whole population in regard to its genetic structure.

The analysis of genetic information with regard to the surnames indicated that the expulsion of the inhabitants of the former Kingdom of Granada and the repopulation of the region were so thorough that it was difficult to note any significant traces of the genetic legacy of the former inhabitants when compared to the genetic North African influence found in the rest of the populations of the Iberian Peninsula.

Autosomal STRs have been widely used in molecular anthropology as an informative ancestry tool for reconstructing human expansion, helping to understand the evolutive history of human populations, and to assess population origins, migrations, and miscegenation. The results of this study illustrate how interdisciplinary collaboration among forensic DNA typing tools such as autosomal STR typing, population genetics analysis, and onomastics may be useful to understand how populations have evolved, sometimes even illuminating obscure episodes in history.

**Supplementary Materials:** The following are available online at https://www.mdpi.com/article/10.3390/genealogy7020029/s1, Figure S1: Geographical distribution of the 245 samples, Table S1: AMOVA design and results from 245 individuals; Table S2: Population pairwise FSTs (above diagonal) and p-values (below diagonal), Table S3: Complete list of populations used in the present study for comparative analysis, Table S4: Evanno table from the 10 replicate runs of Structure calculated with Structure Harvester, Table S5: Observed surnames in the GMA population, Table S6: Allele frequencies of the populations used for Correspondence Analysis for those alleles that make that the Basque Country.

**Author Contributions:** Data curation, M.S.; Investigation, M.S.; Supervision, J.C.A. and J.A.L.; Writing—original draft, M.S.; Writing—review & editing, C.H. and L.J.M.-G. All authors have read and agreed to the published version of the manuscript.

**Funding:** This research received no external funding.

**Institutional Review Board Statement:** This study was approved by the Ethics Committee of the University of Granada (Approval Number: 885). All methods were performed in accordance with the relevant guidelines and reg-ulations of the University of Granada.

**Informed Consent Statement:** All subjects who were recruited for this study gave their informed consent per the Declaration of Helsinki.

**Data Availability Statement:** Data available on request due to restrictions, e.g., privacy or ethical. The data presented in this study are available on request from the corresponding author.

**Acknowledgments:** The authors thank all of the participants who donated buccal swabs and all those who helped in the sample collection—namely, María Luisa Aceituno Villalva, Leticia Olga Rubio Lamia, and Verónica Delgado López. The authors wish to thank M. Bouabdellah's team for providing the Moroccan profiles for the statistical analysis; Andreas Tillmar, Department of Forensic Genetics and Forensic Toxicology, National Board of Forensic Medicine, Sweden for the Swedish profiles; and Carina Schlebusch, Department of Evolutionary Biology, Evolutionary Biology Centre, Uppsala University, Sweden for the southern African profiles. In addition, the authors want to thank Xiomara Gálvez for the technical assistance in the laboratory.

**Conflicts of Interest:** The authors declare no conflict of interest.

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
