# Peer review of "Genetic Population Flows of Southeast Spain Revealed by STR Analysis"

_genealogy, doi:10.3390/genealogy7020029_

Round 1

Reviewer 1 Report

Very good article, involving a subject of great interest and relevance within the area of genetic genealogy. I suggest some structural changes as indicated in the attached file. A revision of figure numbering is required.

Author Response

Response to reviewer 1

Thank you so much for all your comments. Find enclose the answer to your suggestions.

  1. Sections have been included in the Introduction.
  2. I agree with the reviewer that materials and methods should come before Results section. However, the instructions for authors state that the order of the section has to be Introduction, Results, Discussion, Materials and Methods, Conclusions.
  3. Supplementary Table 2 has been included in the manuscript, it was a mistake in the first submission.
  4. All files have been renumbered.
  5. All supplementary files have been included in the submission.
  6. Legend of the Figure 3 has been updated.
  7. Legend of the Figure 4 indicates that they are the surnames from the studied population.
  8. Figures 2a and b are now renamed as Figures 5 a and b.
  9. References from the manufacturer and ISFG recommendations have been included.

Reviewer 2 Report

The article analyzes the presence of genetic influence from North Africa in the population of SE Spain.

Line 37 indicates that Granada was renamed Illiberis during the Roman Empire. Line 41 indicates that when the Berbers arrived it was called Iliberir. What is the correct name?

In the text that appears between lines 42 and 44 it is not clear when the kingdom of Granada was formed. Please clarify it.

Line 73 speaks of a north-south gradient, but after that it speaks of an east-west gradient. Please clarify this text.

The CSF1PO, D18S51, D21S11 and FGA markers, their frequencies do not add up to 1. Please review the data. The tables must be edited in another way. Thus, it is difficult to assess the data.

The table with the AMOVA data is not there.

Authors should also indicate the values Delta of detail in relation Structure.

Line 138 indicates that data from the north of Portugal has been used for the MDS. This point does not appear in the MDS or in table S1. The authors should indicate, in addition to the Stress value, the RSQ value for the MDS.

Can the authors specify more what they mean by AndSPA? Is it a general series of samples from all of Andalusia? Why don't you use the Huelva data that you have used with Structure?

It is not clear what the authors mean between lines 152 and 157. In the correspondence analysis they have separated the Berbers of Asni and Bouhria, and the Arabs. But in Structure they have been mixed with what is logical that it appears that way in Structure. Why these changes in criteria? There must be a reason and it must appear in the article.

It is difficult to understand why the authors choose articles where there are only results for 13 markers when for those populations there are articles with results for 21 markers. For example, the article by Yurrebaso et al 2011 with data for 21 STRs in the population of the Basque Country. They should do a more thorough search.

On line 166 they indicate that the last name most frequently has 23. However, in the graph the last name García exceeds 45%. Do they refer to "different things" when talking about frequency in one case and in another? Please clarify this.

Line 224 refers to data that does not appear in the article. If they are to be raised in the discussion, they must appear in the article.

In line 230 populations from southern Europe and Spain are mentioned again, when there are only Spanish populations.

It is interpreted that the African component has been detected. However, it is not considered that autosomal STRs are not adequate markers to detect the presence of an external genetic component. References to other studies with autosomal SNP markers are not comparable given their mutation rate. Regarding the 2 articles referenced as justification for the use of autosomal STRs, one of them concludes that the results "should be taken with caution because it is based on autosomal STRs with low inter-population variation levels". And the second, ref. 25, uses STRs from the Y chromosome.

In line 287, and in the abstract, it is indicated that there are surnames originating in Castilla or Galicia and "most surnames had higher frequency in southern Spain". The figures do not indicate how frequent these surnames are in the north of the Iberian Peninsula. As it is not indicated, that statement cannot be made.

Table S2 indicates that the Moroccan, Libyan and Somali populations are derived from European populations in the structure analysis. it's right?

Author Response

Response to reviewer 2
Thank you so much for all your comments. Find enclose the answer to your suggestions. The article analyzes the presence of genetic influence from North Africa in the population of SE Spain. Line 37 indicates that Granada was renamed Illiberis during the Roman Empire. Line 41 indicates that when the Berbers arrived it was called Iliberir. What is the correct name? Both names are correct, the city suffered different name changes during the history. In the text that appears between lines 42 and 44 it is not clear when the kingdom of Granada was formed. Please clarify it. The kingdom of Granada suffered different redistributions all over its existence. The first signs of the kingdom were in 711 but it was not until 1090 that the expansion of Al-Andalus was completed is a great territory of the present Iberian Peninsula. However, in 1212 the reign was reduced to the what is actually Granada, Malaga, Almería and some parts of Cordoba, Sevilla, Jaen and Cádiz. Tiny comments have been included in line 44 in order to clarify this issue. Line 73 speaks of a north-south gradient, but after that it speaks of an east-west gradient. Please clarify this text. What it really wants to say in the text is that contrary what it might be expected based on historical data that points to a south-north increasing gradients; genetic data indicates an east-west gradient. Maybe commas were not properly situated. I hope it is clear now. The CSF1PO, D18S51, D21S11 and FGA markers, their frequencies do not add up to 1. Please review the data. The tables must be edited in another way. Thus, it is difficult to assess the data. All allelic frequencies have been recalculated and the data has been updated. This table is designed to be published horizontally. I will make sure with the editing team that in the final version of the paper the table is properly oriented. The table with the AMOVA data is not there. I am extremely sorry but I do not know why supplementary data was not included in the revisions. Authors should also indicate the values Delta of detail in relation Structure. Delta data has been included in the manuscript for K=5 and detail of all the run has been included as Supplementary Table 3. Line 138 indicates that data from the north of Portugal has been used for the MDS. This point does not appear in the MDS or in table S1. The authors should indicate, in addition to the Stress value, the RSQ value for the MDS. It has been a mistake; this population was in the first analysis but eliminated latter. Moreover, RSQ value has been included. Can the authors specify more what they mean by AndSPA? Is it a general series of samples from all of Andalusia? Yes, it is a general set of samples from the whole andalusian community. Why don't you use the Huelva data that you have used with Structure? Data from AndSPA from genetic distance and correspondence analysis come from allelic frequency data whereas data from Huelva for Structure analyses come from genetic profiles. As it was not possible to get the genetic profiles from the whole Andalusian population, we had to select the population where we could get the genetic profiles for the structure analysis. It is not clear what the authors mean between lines 152 and 157. In the correspondence analysis they have separated the Berbers of Asni and Bouhria, and the Arabs. But in Structure they have been mixed with what is logical that it appears that way in Structure. Why these changes in criteria? There must be a reason and it must appear in the article. Data from Berbers of Asni, Bouhria and Arabs come from allelic frequency data and data from Moroccan population for Structure analyses come from genetic profiles. It was impossible to get
the genetic profiles from Berbers of Asni, Bouhria and Arabs. That is why different populations are used for the different analysis. Supplementary Table 1 contains the populations used in each type of analysis, number of individual, number of markers and references. It is difficult to understand why the authors choose articles where there are only results for 13 markers when for those populations there are articles with results for 21 markers. For example, the article by Yurrebaso et al 2011 with data for 21 STRs in the population of the Basque Country. They should do a more thorough search. For some of the populations; as Turkey, Catalonia or Andalusia, there was only genetic data available from 13 markers. That is why the comparisons were reduced to 13 markers. It is true that for some of the populations included in the study there is information for more than 13 markers but as for not all the populations included this information is available, we had to reduce the number of markers used. On line 166 they indicate that the last name most frequently has 23. However, in the graph the last name García exceeds 45%. Do they refer to "different things" when talking about frequency in one case and in another? Please clarify this. It is true that it can be misunderstood. In fact, 23 is the number of times the most frequent last name has been detected in the GMA population, so it is the absolute frequency. 46.37‰ is the relative frequency. Line 224 refers to data that does not appear in the article. If they are to be raised in the discussion, they must appear in the article. In line 123 it is stated “To simplify the interpretation of the data, the figure omits markers’ data and shows only population results.” That is why, in the discussion there is reference of data from alleles that are not reflected in the figure. The inclusion of alleles in correspondence analysis diagrams, makes them extremely confusing and it has been only used for discussion purposes. In line 230 populations from southern Europe and Spain are mentioned again, when there are only Spanish populations. Southern Europe population has been eliminated. It is interpreted that the African component has been detected. However, it is not considered that autosomal STRs are not adequate markers to detect the presence of an external genetic component. References to other studies with autosomal SNP markers are not comparable given their mutation rate. Regarding the 2 articles referenced as justification for the use of autosomal STRs, one of them concludes that the results "should be taken with caution because it is based on autosomal STRs with low inter-population variation levels". And the second, ref. 25, uses STRs from the Y chromosome. This paper is part of a bigger study were not only autosomal but also Y-chromosome STRs and mitochondrial DNA were evaluated to asses the possible African contribution to present population of the former Kingdom of Granada. The reference 25 has been updated to an actual reference of autosomal STR. In line 287, and in the abstract, it is indicated that there are surnames originating in Castilla or Galicia and "most surnames had higher frequency in southern Spain". The figures do not indicate how frequent these surnames are in the north of the Iberian Peninsula. As it is not indicated, that statement cannot be made. It is true that the figures do not reflect the relationship between the region of origin of the surnames and the regions with higher number of individuals born with each surname. References to the figures have been eliminated in line 287. In order to answer your question. i.e. last name Padial, according to historical records, was originated in Galicia. However, nowadays this last name has 0 representation in Galicia and the province where the higher frequencies detected for this last name is Granada, 11.04%. I have indicated this data in the paper but I am not sure if the reviewer also wants the data of all the 61 last names that were originated in a region different to the one where is most frequent to be reflected in the paper. Table S2 indicates that the Moroccan, Libyan and Somali populations are derived from European populations in the structure analysis. it's right?
I am extremely sorry, it has been a mistake that has been corrected in the resubmission.

Round 2

Reviewer 2 Report

We thank the authors for the effort they have made to clarify the text. Just a few comments.

*In the answer the authors specify that in 1212 the kingdom was reduced to what today corresponds to Granada, Almería, Malaga and some parts of Corboda, Seville, Jaen and Cadiz. This information is important and should appear in the text.

*In the AMOVA table a value of 2.32% appears but in the text (line 114) 2.23% appears.

*In relation to the data that does not appear in the graphs and that are discussed as the allele frequencies of certain markers, the authors can present tables, only of the markers D13S317, TPOX, and TH01 with the allele frequencies of the analyzed populations. Otherwise, that paragraph should not be in the discussion since the reader cannot assess the data.

*As the authors indicate in their letter, studies with other lineage polymorphisms such as mitochondrial DNA or Y chromosome polymorphisms are necessary. The authors should make some mention of this, or else support their argument with data on these polymorphisms from other works given the low resolution of autosomal markers. Precisely in reference 25 they reach conclusions opposite to those of this article, highlighting the proximity of the Berber-speaking population of Souss with the populations of Andalusia.

*The authors should end the paper with a Conclusions section.

Author Response

We thank the authors for the effort they have made to clarify the text. Just a few comments.

We thank the reviewer the extensive work and all the comments.

*In the answer the authors specify that in 1212 the kingdom was reduced to what today corresponds to Granada, Almería, Malaga and some parts of Corboda, Seville, Jaen and Cadiz. This information is important and should appear in the text.

The information has been included in the text.

*In the AMOVA table a value of 2.32% appears but in the text (line 114) 2.23% appears.

It has been a typographical error, the correct value is 2.32%

*In relation to the data that does not appear in the graphs and that are discussed as the allele frequencies of certain markers, the authors can present tables, only of the markers D13S317, TPOX, and TH01 with the allele frequencies of the analyzed populations. Otherwise, that paragraph should not be in the discussion since the reader cannot assess the data.

A new supplementary Table has been included with this information. 

*As the authors indicate in their letter, studies with other lineage polymorphisms such as mitochondrial DNA or Y chromosome polymorphisms are necessary. The authors should make some mention of this, or else support their argument with data on these polymorphisms from other works given the low resolution of autosomal markers. Precisely in reference 25 they reach conclusions opposite to those of this article, highlighting the proximity of the Berber-speaking population of Souss with the populations of Andalusia.

A statement with this observation has been included. Furthermore, in line 261, there is a reference to our previous study with Y-STRs.

In reference 25, the Andalusian population is a population from Huelva and analyzing the data from genetic distances, there are no bigger differences between the “Andalusians” and Souss (0.040) rather than with the Spanish population (0.035).

The references used in reference 25 to indicate influence on the Andalusians “Conversely, other study conducted on the Northwest African population (Bosch et al., 2000) has reported a genetic discontinuity between the Northwest African and the Iberian populations, with the exception of the Andalusians, who seem to be influenced by a northwest African gene flow.” indicates that “A clear genetic difference was found between NW African populations and Iberians, which underscores the Gilbraltar Straits as a strong barrier to genetic exchange; nonetheless, some degree of gene flow into Southern Iberia may have existed”.

On the other hand, they state that “Macedonians (MK) and Andalusians (AD) show more affinity with the Souss population with only one significant difference.” but in the conclusions they state “ we observed that the Souss population had a close genetic affinity with south Europeans and Arabs, confirming the historical connections between these populations”. In my opinion, they comment their Fst results, that indicate that among the South European populations, the Andalusians and the Macedonians are the ones that only show one significant difference with Souss population but finally they do not conclude any more relevant data about these findings.

*The authors should end the paper with a Conclusions section.

The conclusion section is located after the materials and methods section, point number 5.